

# The first global 883 GHz cloud ice survey: IceCube Level 1 data calibration, processing and analysis

Jie Gong[1,2], Dong L. Wu[2], and Patrick Eriksson[3]

[1]GESTAR/USRA, Columbia, MD, United States
[2]NASA Goddard Space Flight Center, Greenbelt, MD, United States
[3]Chalmers University of Technology, Gothenburg, Sweden

**Correspondence:** Jie Gong (Jie.Gong@nasa.gov)

**Abstract.**

Sub-millimeter (sub-mm, 200 - 1000 GHz) wavelengths contribute a unique capability to fill-in the sensitivity gap between operational visible/infrared (VIS/IR) and microwave (MW) remote sensing for atmosphere cloud ice and snow. Being able of penetrating cloud to measure cloud ice mass and microphysical properties in the middle to upper troposphere, this is a critical spectrum range for us to understand the connection between cloud ice and precipitation processes.

As the first space-borne 883 GHz radiometer, IceCube mission was NASA's latest effort in spaceflight demonstration of a commercial sub-mm radiometer technology. Successfully launched from the International Space Station, IceCube is essentially a free-running radiometer and collected valuable 15-month measurements of atmosphere and cloud ice. This paper describes the detailed procedures for Level 1 data calibration, processing and validation. The scientific quality and values of IceCube data are then discussed, including radiative transfer model validation and evaluation, as well as the unique spatial distribution and diurnal cycle of cloud ice that are revealed for the first time on a quasi-global scale at this frequency.

## 1 Introduction

Ice clouds play crucial roles in Earth's climate and weather through interactions with atmospheric radiation, dynamics and precipitation processes at a wide range of spatial and temporal scales [Wu et al. (2019)]. By far ice clouds remain the leading source of uncertainties in weather model predictions and future climate projections [Stocker et al. (2013)] mainly because of two reasons. On one hand, upper-tropospheric ice clouds are radiatively important to the Earth's energy budget, and inaccurate measurements of ice cloud properties add poor or even misleading constraints on ice cloud radiative feedback (CRF) in models. On the other hand, large ice crystals fall and form precipitation eventually. Traditional remote sensing techniques leave gaps in this cloud-precipitation transition and coupling process. As a result, ice cloud and associated radiative and hydrological properties simulated by GCMs vary wildly and are often subjected to heavy tuning to close the radiation budget at the top of the atmosphere and precipitation at the surface, leaving the middle of the process highly uncertainty and under constrained [Waliser et al. (2009); Li et al. (2016)].

Ice clouds net radiative effect can be positive or negative depending on its macrophysical (e.g., cloud top height, optical thickness) and microphysical (e.g., particle size distribution, particle shape, etc.) properties. By far large discrepancies still





remain among various satellite ice cloud retrieval products, and between observations and reanalyses and/or GCM simulations. Traditional remote sensing techniques working at visible (VIS) and infrared (IR))spectrum ranges are sensitive to ice cloud properties such like cloud top height, optical thickness and particle size, etc [Wang et al. (2016)], but suffer from only being able to penetrate the top few kilometers of the cloud layer [Eliasson et al. (2011)]. For example, Eliasson et al. (2011) and Duncan and Eriksson (2018) identified more than 300% discrepancies among ice water path (IWP) retrievals derived from

different A-Train satellite instruments, albeit they all sample the same body of cloud at the same local time.

Moreover, achieving consensus about ice cloud mass is unavoidably a critical step when attempting to close the global hydrological budget. Unfortunately the aforementioned sensitivity gap between VIS/IR and MW results in the missing piece of inter-coupled process that connects atmospheric cloud with precipitation-sized hydrometors. Although combined spaceborne radar and lidar observations (e.g., DARDAR or 2C-ICE products) largely alleviated this problem, they are not suitable for

capturing daily weather-scale variabilities because of narrow swaths and in-frequent revisit time. Relatively inexpensive passive sensors are still required to cover the wide spatial and temporal scales of weather as well as to produce the climate record.

The sensitivity gap can be visually quantified from Fig. 1a. In this figure, the 2-dimensional probability density functions (PDFs) between $T_{cir}$ and $IWP$ are constructed based on pure observations from a passive infrared sensor (Atmospheric Infrared Sounder, or AIRS) and a passive microwave sensor (Microwave Humidity Sounder, or MHS), respectively. $T_{cir}$, so-

called "cloud-induced radiance depression", is defined as the difference between the observed brightness temperature ($TB$) and the simulated clear-sky radiance ($T_{ccr}$) by removing any frozen hydrometeors from the column (Gong and Wu (2014), Wu et al. (2019)). $IWP$ "truth" is extracted from the joint spaceborne radar and lidar product called 2C-ICE, and $T_{cir}$ values are taken from collocated IR and MW observations, respectively. From Fig. 1a we can see the sensitivity range, where the slope is steep enough to be eye-balled, is about $10-70g/m^2$ for passive IR sensors; however, $T_{cir}$ for a typical passive MW sensor

does not start to become sensitive to IWP before it exceeds $300g/m^2$ or so, as marked by the horizontal rectangles in Fig. 1a. The sensitivity range for IceCube, based on a radiative transfer model simulation (blue curve), is between 40 and $600g/m^2$, which perfectly fills in the gap between IR and MW sensors (orange rectangles). Readers are directed to the Appendix A for details of observational datasets, collocation algorithm and model simulations. Hence, we add the "sub-mm" sensitivity column (blue cylinder) in Fig. 1 of Eliasson et al. (2011), as shown here in Fig. 1b. Admittedly IR+VIS (yellow cylinder) could largely

cover the same sensitivity range of sub-mm, but visible spectrum loses its capability at nighttime and polar night season, and furthermore suffers from multiple scattering to small ice crystals.

The $874GHz$ channel ( or more generally speaking, $860-900GHz$ range) not only provides great sensitivity to cloud ice scattering, but also allow sufficient penetration to measure upper-middle tropospheric cloud ice mass and size properties. Gruntzun et al. (2018) found out that the sensitivity level to cloud (snow) ice peaks at $\sim 300hPa$ ($400hPa$) for $874GHz$

at the nadir-view using the Atmospheric Radiative Transfer Simulator (ARTS) and its comprehensive ice scattering database and absorption spectrum (Buehler et al. (2018)) . This channel also has a strong positive response at about $200-300hPa$ to water vapor, meaning that it can be merely contaminated by complicated surface signals or liquid/rain cloud in the lower troposphere, unless the upper-troposphere is extremely dry. Since it is relatively easy to obtain water vapor information in the upper-troposphere (e.g., Moradi et al. (2015), Bosilovich et al. (2017)), $874GHz$ is an ideal channel for retrieving ice cloud



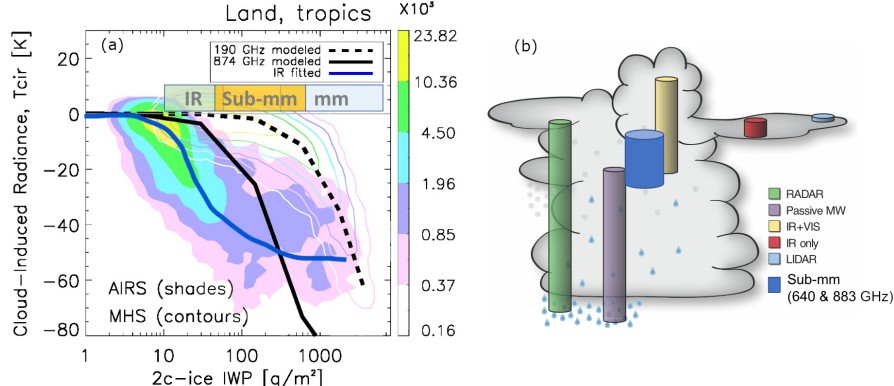

**Figure 1.** (a) 2-dimensional probability density functions (PDFs) of cloud induced radiance perturbation ($T_{cir}$) and ice water path (IWP) derived from collocated CloudSat/CALIPSO and AIRS (color shaded) and MHS (color contours) observations in the tropics ($30°S - 30°N$). The blue solid line connects the PDF peak for AIRS-CloudSat/CALIPSO. Black solid (dashed) line is from the ARTS model simulation for 874 (190) GHz. See the Appendix 1 for details. (b) a conceptual depiction of the rough penetration depth of different passive and active spaceborne sensors. This panel is modified from Fig. 1 in Eliasson et al. (2011) by adding the sub-mm sensitivity column (blue).

mass for medium thick ice clouds located above $400hPa$. Moreover, radiative transfer model (RTM) simulations also indicate this frequency is most sensitive to ice particles with effective diameter ($D_e$) between $100 - 200\mu m$ (Buehler et al. (2007)). While numerous literatures have revealed that MW Mie scattering only happens for precipitation-size particles ($D_e > 200\mu m$), VIS channel sensitivity range is $10 - 120\mu m$ (Platnick et al. (2017)), and IR sensitivity range is $20 - 100\mu m$ (Garnier et al. (2013), Wang et al. (2016)), $874GHz$ not only fills in the sensitivity gap of ice cloud mass, but also the gap of ice particle

size spectrum. Above that frequency at THz band, cloud ice remote sensing becomes more difficult because of increasing atmospheric attenuation from gas continuum absorption (Wu et al. (2019)).

Working at $883GHz$ with the lower sideband at $874GHz$, IceCube is one of the only three known instruments that were ever built to work at this frequency. Therefore, making IceCube data available to the public is of critical importance to benefit the entire scientific community to advance our understanding of ice cloud radiative feedback and cloud-precipitation coupling

processes.

This paper is organized as follows. Section 2 describes the procedures for data calibration, gain model construction and geolocation registration. Section 3 validate the quality of the delivered Level 1 radiance data against other satellite datasets, previous campaign data collections as well as RTM simulations. Section 4 discusses some scientific applications of this dataset. Section 5 includes the conclusion and some discussions about future directions for instrument design and orbital selection.

## 2   Procedures for Level 1 data calibration and processing

IceCube mission was funded by NASA to make a fast-track spaceflight demonstration and validation of the Virginia Diodes, Inc (VDI)'s $883GHz$ receiver for cloud ice observations. VDI's commercialized $874GHz$ receiver greatly reduced the cost of





this mission, yet the stability of this receiver in space is uncertain. IceCube was successfully deployed from the International Space Station (ISS) on May 16, 2017, and reentered the Earth's atmosphere on Oct 2, 2018. During its $\sim 15$ months of life
in space, IceCube not only successfully completed its primary goal to retire the risks for VDI receiver, it also collected great amount of scientifically-valuable data over the low-mid latitudes that lead the first ever global cloud ice map at this frequency. This Section dedicates to describing the procedures we carried out to deliver the Level 1 radiance data (Gong and Wu (2021)).

To reduce the mission risk and power usage, IceCube was chosen to fly without a scan mirror for radiometric calibration. Instead, it spins the spacecraft to obtain periodic views between cold space and Earth's atmosphere. Thus, the cold space count
measurements and clear-sky Earth atmosphere are used for the radiometric calibration and receiver gain calculation. This simplified engineering design successfully kept the cost low, but poses some challenges for the data calibration, geolocation registration and data processing. IceCube antenna produces a $1.8°$ half-power beam width, which translates into a $12.6km$ nadir footprint size at ISS orbit ($\sim 400km$), which gradually decreased as IceCube spacecraft dropped down its orbit height. Interested readers are encouraged to read Wu et al. (2019) for more details regarding instrument design, altitude control, etc..

## 2.1   Cold space calibration and offset

IceCube is essentially a free-running radiometer. As it lacks a stable cold/hot reference for absolute calibration, IceCube is calibrated against the predicted "space counts" in each spin, which should be calibrated to 0 all the time. Here "count" stands for the digitized number of voltage. The space count ($C_{sp}$) is not a constant, but a strong function of the receiver temperature ($Tp$) and the relative time duration with respect the most recent switch-on time ($dt$). A noise source injector is included so the
output voltage counts have four modes: antenna plus noise (Ant+N), antenna (Ant), reference (Ref), and reference plus noise (Ref+N) . Adding noise is for testing instrument calibration function, so the measured total voltage count ($C$) is defined as $C_{Ant} - C_{Ref}$, where $C$ is a function of $Tp$, as shown in Eqn. 1:

$$C = C_e + C_{sp}(Tp, dt) = G(Tp, \Delta t) \cdot T_b + C_{sp}(Tp, dt) + R_{sp} \tag{1}$$

Here $C_e$ is the Earth-view counts. $G$ is the instrument gain, which is also a function of $Tp$ and $\Delta t$. The latter corresponds
to the Julian day counted from January 1st, 2017, as instrument gain is degrading slowly through time, but is considered stable in each day. $T_b$ is the Level 1 radiance that we aim to get at the end. $R_{sp}$ is the residual of $C_{sp}$ which could not be fitted by our space count fitting procedures. Four temperature readings were recorded, which are temperatures for the isolator ($Tp_1$), detector ($Tp_2$), reflector ($Tp_3$) and mixer ($Tp_4$). They are highly correlated, but as the mixer is closest to the output end, we use this record to represent $Tp$ for the first two steps of calibration. Other three $Tp$s are used subsequently in the machine
learning/artificial intelligence (ML/AI) training step to identify any trivial contributions to the unexplained residual.

To give an example, Fig.2a plots the time series (with respect to the orbit switch-on time) of the raw counts ($C$) measurements. The first step shown in Fig.2b is to apply a 2nd order polynomial fitting twice with respect to $Tp_4$ to remove the temperature-dependent variation of $C_{sp}$, which is successfully captured (red line). The second step is to remove the periodically varying component by a 5th order polynomial fitting (blue line), which may associate with IceCube location and/or spin
velocity. The third step is to further remove the periodically varying component with respect to $Tp_3$ and $Tp_4$ by applying a



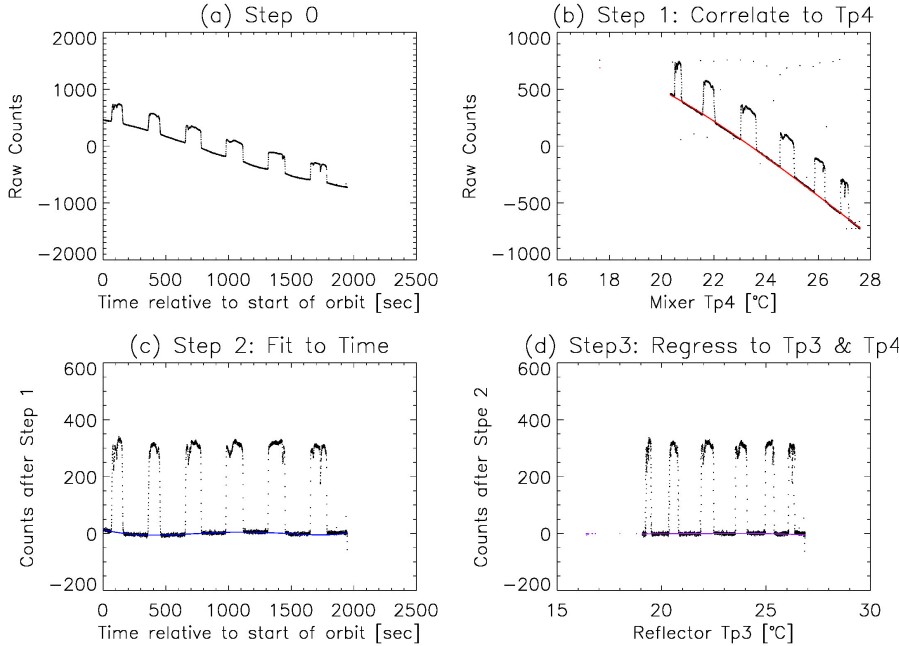

**Figure 2.** An example showing the four steps of relative calibration: (a) raw total counts ($C$); (b) raw counts polynomial fitting to $Tp_4$ (red) can largely capture the $C_{sp}$ variation; (c) count residuals polynomial fitting to relative time (blue); (d) count residuals polynomial fitting to $Tp_3$ and $Tp_4$ (purple). After these four steps, standard deviation of $C_{sp}$ decreased to $\sim 4$ counts. This orbit is taken from the first orbit of IceCube on August 14, 2017.

sinusoidal fitting to $Tp_3$ and then $Tp_4$ (purple line), respectively. After removing this low-frequency component, the residual of $C_{sp}$ is about 4 counts for this case, and the Earth-view counts $C_e$ now remain stable on this orbit, and are clearly separated from the $C_{sp}$. The 5th step is to separate them by detecting the sudden jumps along the time series, as shown by the blue and red crosses in Fig. 3. Once the jumping points are identified, the average time in-between is considered as the "nadir-view" time (indicated by thin black vertical lines). Now we can see "dips" in some segments of $C_e$ in Fig. 3, e.g., the one at $dt = 710s$ and the one at $dt = 1720s$. These "dips" are induced by cloud scattering.

However, it turned out that Step 3 and Step 4 may not remove all slowly varying $C_{sp}$ component, as suggested by a "bad example" in the Appendix B. Therefore, a further polynomial fit is carried out (green thin line in Fig. 3) to secure the stability of $C_e$. In this final step, the spin velocity is calculated and compared to the recorded angular velocity. If the spin velocity is too slow, or if the contrast between adjacent $C_e$ and $C_{sp}$ is too small, this spin is considered of poor quality and is entirely excluded for the final Level 1 product (see the Appendix B for examples).

The residual ($R_{sp}$) (i.e., difference of the green dots and the green line on Fig. 3) provides a good estimate of the noise level of $C_e$, which remains $\sim 4K$ throughout the mission (black crosses in Fig. 4). Since only $dt$, $Tp_3$ and $Tp_4$ are used for the empirical calibration steps 1-5, a ML/AI model was trained and used as the final step to check whether the noise ($\sigma_{sp}$) can be

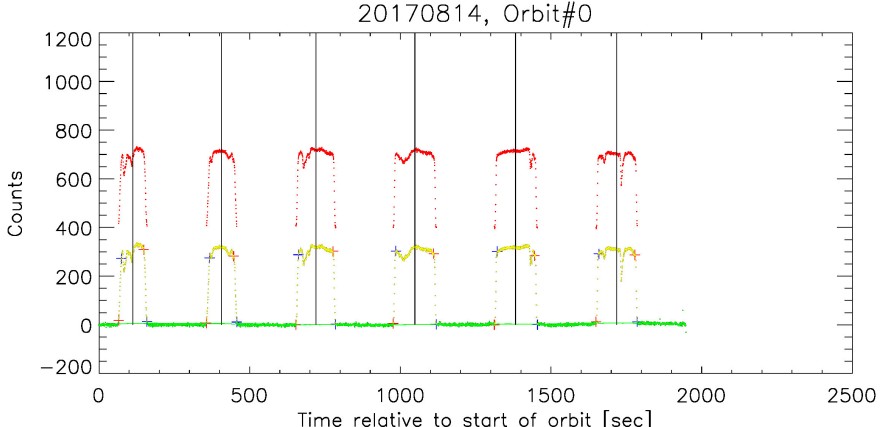

**Figure 3.** Further separation of space-view (green) and Earth view (yellow) by detecting the jumping thresholds (blue and red crosses). After separating the views, $C_{sp}$ was further fitted again by a polynomial fitting to remove slowly varying component. The red dots (shifted upward by 200 counts) are the final Earth voltage counts that will be converted to $TB$. This final polynomial fitting seems redundant here for a good orbit, but is quite necessary for a bad orbit, an example of which is given in the Appendix B. The average time between the starting and ending times of each Earth-view spin is assigned as the "nadir-view" time (thin black vertical lines).

further reduced. Here $\sigma_{sp}$ is defined as the standard deviation of the residual $R_{sp}$. After applying this model, $\sigma_{sp}$ in deed was reduced to $\sim 2K$ as shown by the red crosses in Fig. 4. While details of the ML/AI model can be found in the Appendix C, this exercise is somewhat illuminating to the engineering team: ML/AI may serve as a general and cost-effective calibration approach for any instrument to combat after-deployment variations that are not well understood in the pre-launch phase, and can be relatively easily carried over to a constellation of cubesats/microsats.

## 2.2    View-angle and geolocation registration

IceCube essentially flies on the same orbit with ISS. However, due to its free-spinner design, its latitude coverage can reach up to $58°N/S$. Spin rate is a critical parameter in order to accurately register the geolocation of an observation. Spin rate in the unit of degrees per second (dps) is the second tier variable to determine. In addition, because geolocation is necessary for the calculation of clear-sky radiance $T_{ccr}$, which is used for constructing the gain model as well as for the IWP retrieval, spin rate
is subsequently a must.

IceCube has three spinning modes. In the daytime, it spun around the sun vector (-Y axis) at Sun point (SP) mode (-1 dps) or Fine Reference Point (FRP) mode (-1.2 dps). At nighttime, it spun around the geomagnetic field (+Z axis) at a speed between +1 to +2 dps. Three axis spin rates are recorded down as $spin_x$, $spin_y$ and $spin_z$. However, the observed spin rate does not necessarily agree with the recorded numbers. As a matter of fact, a systematic low-bias was identified when the view-angle is greater than $30°$ at the SP mode (see Fig. 1-10 in Wu et al. (2019) for the comparison). We therefore threw away all observations with view-angle beyond $50°$, and assign a low-quality flag to those between $30°$ and $50°$.




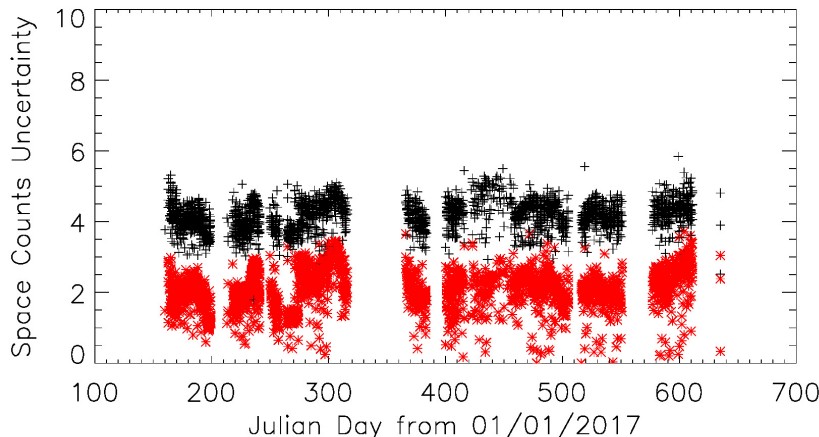

**Figure 4.** Time series of $\sigma_{sp}$ before the ML/AI model fitting (black crosses) and after the fitting (red crosses). Each cross represent one orbit.

After we obtain the Earth-view $C_e$ and identified the limb-to-limb time (LLT) and nadir-to-nadir time (NNT), the ratio between measured and calculated LLT and NNT are inspected, respectively. In Fig. 3, LLT is represented by the time difference between each pair of the blue and red crosses, and NNT is represented by the time difference between two adjacent vertical
lines. We found the ratio from measured and calculated LLT for daytime is correlated with the beta-angle variation, while LLT ratio at nighttime and NNT ratio at both daytime and nighttime remain rather stable, varying between 0.9 to 1.1 (see Fig. 1-11 in Wu et al. (2019)). We hence use NNT ratio as a scaling factor to multiply the measured spin rate, which is then used for view-angle calculation and geolocation registration (with the altitude and orbit information well-known from the orbital TLE parameters). If this ratio is beyond 1.1 or less than 0.9, this spin is excluded in the subsequent processing (e.g., the last 5 spins
in Fig. B2 are excluded due to slow spin rate). Overall there is estimated of $\sim 10\%$ uncertainty associated with the view-angle, which induces up to 1 footprint offset (or $0.1°$ latitude/longitude offset) for geolocation registration using the aforementioned processes. Readers are referred to Section 1.4.1 of Wu et al. (2019) for details of this part.

## 2.3 Gain model reconstruction

The gain model $G(Tp, \Delta t)$ is the ratio between $C_e$ and the Level 1 radiance or brightness temperature $TB$. It is a function
of instrument $Tp$ and days in the orbit to account for the instrument degradation. As ice cloud scattering strongly depresses the Earth-view voltage counts $C_e$, simulated clear-sky $T_{ccr}$ is used to compute the ratio and to construct the gain model. $T_{ccr}$ is computed from the radiative transfer model (RTM) used for the Aura-Microwave Limb Sounder (Wu et al. (2006)). In this RTM, dry and wet continua, as well as line emissions including broadening parameters for ozone lines were computed following Wu and Jiang (2004). This RTM requires input of atmospheric profiles, including temperature, water vapor and
ozone interpolated from the MERRA-2 reanalysis (Bosilovich et al. (2017)) using the nearest neighborhood method. In addi-



tion, footprint geolocation, timing and view-angle (or solar zenith angle) are also required. In practice, given the fact that the nadir-view footprint location and timing are well determined from the orbital TLE parameters, we can then compute the $360°$ full-spin radiances using three spin rate assumptions: 0.8 dps, 1 dps and 1.2 dps. $T_{ccr}$ is then interpolated (or extrapolated) given the actual spin rate and view-angle.

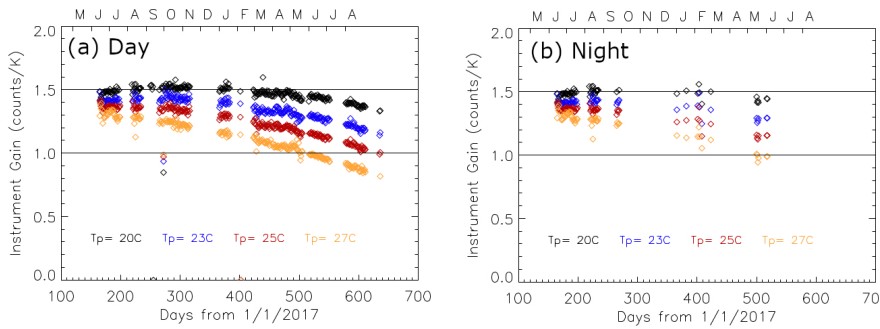

**Figure 5.** Time series of daily gain for (a) daytime and (b) nighttime when $Tp_4$ value equals to $20°C$ (black), $23°C$ (blue), $25°C$ (red), and $27°C$ (orange). The gain declines consistently with increasing of instrument temperature.

Gain was computed on a daily basis from the ratio of $C_e$ and $T_{ccr}$ as a function of $Tp_4$. Everyday we can compute the 2D statistics from the global samples and identify the most probable $Tp_4$ - $G$ relationship. A second order polynomial fitting was then applied to parameterize this relationship so to construct the final look-up-table (LUT) as shown in Fig. 6. The daily variation of $G$ are plotted in Fig. 5, separately for day and night. We can see although the night yield of high-quality observations are fewer than those during the daytime because IceCube was turned off during most of the night to save power, the magnitude

of gain and its degradation overtime are almost identical. Therefore, we don't separate day and night in the final construction of the LUT. Yet small variations of the gain can be visually identified from Fig. 5, which is nevertheless omitted with using the LUT in Fig. 6. But in rare situations when the daily gain dropped dramatically (e.g., the outlier at ∼Day 270 in Fig. 5), data collected for the abnormal day were excluded completely. It's worth mentioning the history of the gain. It was measured at 2.37 count/K at $20°C$ during the instrument thermal vacuum test (TVAC). However, it dropped to 1.1count/K at $20°C$ after

the integration and testing (I&T) phase, which was suspected to be caused by debris falling into the unprotected receiver's freehorn during the I&T phase. Nevertheless, the in-flight gain at $20°C$ (black circles in Fig. 5) never dropped to below 1.3 count/K fortunately. $C_e$ is converted to $TB$ using the gain LUT shown in Fig. 6. Furthermore, $TB$ values are flagged as "poor quality" when the gain is smaller than 0.9.

     As the final step, $TB$ is computed from Eqn.1. Note that we can also estimate the uncertainty of $TB$ at the same time, which

is $2 - 4K$ for most orbits. Details of $TB$ uncertainty estimation can be found in the Appendix D.



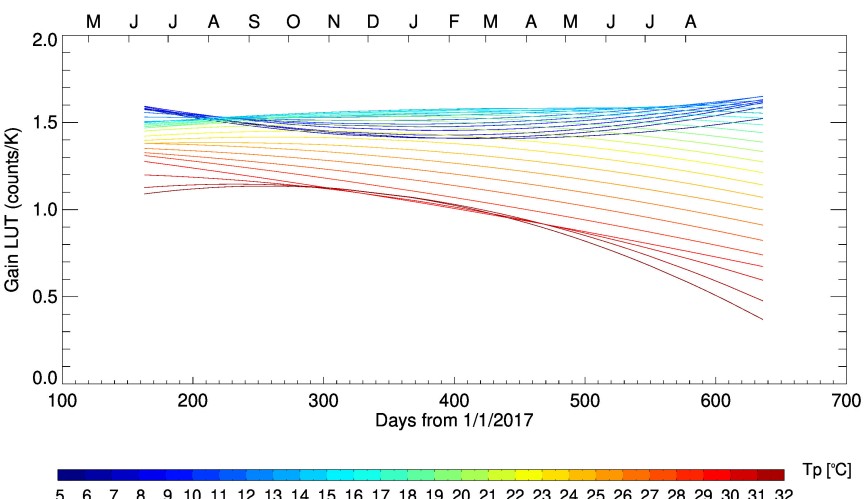

**Figure 6.** Look-up table (LUT) of the gain model as a function of $Tp_4$ and Julian day.

## 3  Level 1 data analysis

### 3.1  Comparison against model simulations

To evaluate the quality of IceCube Level 1 $TB$, we will first compare the general statistics of $TB$ against RTM simulations. The RTM employed here is the ARTS model. This model is chosen not only because it is one of the very few RTMs that can conduct cloud simulations at 874-883 GHz, but also because of its well-developed comprehensive ice scattering databases and absorption spectrum at sub-mm frequencies (Buehler et al. (2018)). CloudSat retrieved IWC profiles together with ECMWF-Interim atmospheric profiles in the deep tropics ($15°S - 15°N$) during June, July and August, 2017 are used as the input parameters for ARTS model simulations, because 883 GHz is the least impacted by surface emissivity contaminations in the humid tropics, and IceCube data quality was the best in these three months of flight. To limit the "beam-filling" effect, this comparison is only limited to IceCube observations within $\pm 15°$ view-angles.

CloudSat carries a spaceborne 94 GHz nadir-pointing Cloud Profiling Radar(CPR) orbiting on a sun-synchronized afternoon polar orbit. The synthetic 883 GHz $TB$ is generated using the "dBZ-based" method described in details in Ekelund et al. (2020). Briefly speaking, given a particle size distribution (PSD) and a particle shape model (PM), radar backscattering cross-section from ice and/or liquid can be computed and associated with a CloudSat radar reflectivity. An onion-peeling method is then applied from the cloud top down layer-by-layer to account for the two-way radar attenuation in each layer by IWC or rain water content (RWC) as well as multiple-scattering. The IWC and RWC grids are calculated iteratively downward to construct the final profile. Then the retrieved IWC and RWC vertical profiles as well as ECMWF-Interim atmosphere gas and water vapor are supplied to the forward model ARTS to compute the synthesized 883 GHz $TB$ using IceCube channel spec. Through



Earth System
Science
Data

this "dBZ-based" way, the consistency is kept for microphysics assumptions for hydrometeors between CloudSat radar and
IceCube radiometer.

Two particle PSDs, namely MH97 (McFarquhar and Heymsfield (1997)) and F07T (Field et al. (2007), tropics), are
considered. MH97 has been used widely in the sub-limb and sub-mm community (e.g., Wu et al. (2006); Eriksson et al.
(2007)). In contrast to many other PSDs, MH97 uses the volume-equivalent (or mass-equivalent) diameter for the size ($D_{me}$),
rather than using the optically-defined effective diameter ($D_e$), so the PSD-integrated mass is not dependent on density and
is hence mass conserved with respect to different PMs. F07 is a single-moment PSD based on in-situ data collected from
multiple campaigns, and has different settings for tropics (F07T) and mid-latitude (F07M). Similar to the gamma-size PSD,
it is not mass conserved. F07 has been used widely in the passive microwave community (e.g., Kulie et al. (2010), Geer and
Baordo (2014)) and are believed to have better representation of precipitation-sized frozen particles while MH97 is believed
to represent anvil or cirrus ice better (a good comparison is given in Fig. 2 of Ekelund et al. (2020)). 8 particle shape models
are tested for each PSD assumption, as listed in the legend of Fig. 7. Random orientation is assumed for all simulations. More
descriptions of model set-ups, including sensitivity to PSD and PM assumptions, can be found in Ekelund et al. (2020).

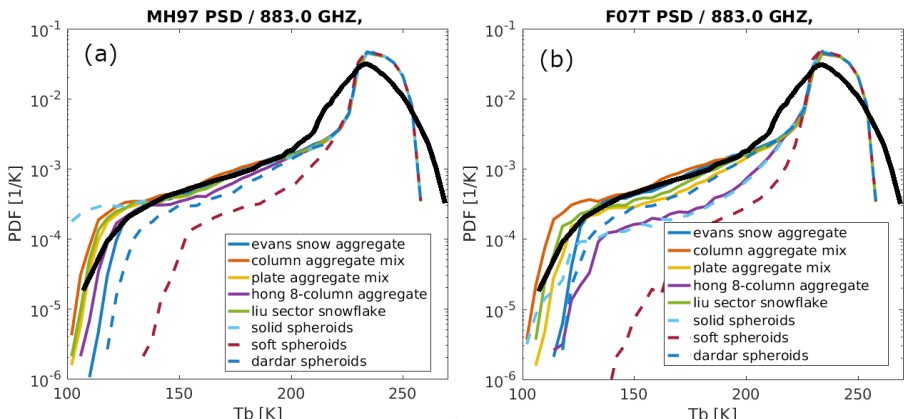

**Figure 7.** PDF comparison of the IceCube TB (black bold line) against various ARTS model simulations using 3 months of tropical CloudSat
observations and different ice shape assumptions and two particle size distribution assumptions: (left) MH97 PSD; (right) F07 PSD. Only
IceCube observations within $\pm 15°$ view-angle, $15°S - 15°N$, and June-July-August, 2017 are considered.

Based on the PDF comparison of simulated and observed $TB$ shown in Fig. 7, we can clearly see that model simulations
capture well the peak of the IceCube observed radiances, although the clear-sky variation (roughly for $TB > 220K$) is sig-
nificantly smaller than the observed. If a Gaussian noise with width of $7K$ is added to the ARTS simulation, it can greatly
mimic the IceCube statistics (not shown), we thus speculate that the warm-end discrepancy is likely induced by the noiser
IceCube data. This Gaussian-shape of random noise mainly originates from three components: Gain uncertainty induced error,
space counts prediction induced error and geolocation registration induced error. The former two factors are explained in more
detail in the Appendix D, which is $2 - 4K$. The 3rd factor's contribution is therefore estimated to be $\sim 3K$. Nevertheless, we
cannot completely exclude other factors, e.g., water continuum might be slightly over-represented in the model. At the cold





end however, model-observation discrepancies reveal some interesting microphysical information. Firstly, none of the spheroid PM assumptions could make cold enough $TB$s that IceCube observe (except solid spheroid with MH97 assumption), indicating that current sphere or spheroid assumptions in many retrieval algorithms and model microphysical schemes are probably wrong or at least not sufficient to capture the observed cloud ice variabilities and radiative properties. Furthermore, even for non-spheroid particles, the differences among produced PDFs are discernible as well. For example, the "Hong 8-column aggregate" (purple solid) seems to well capture the observed statistics with the MH97 assumption, but apparently produces too warm $TB$ with the F07T assumption. This particle shape is very similar to the PM assumption used in the current MODIS cloud retrieval algorithms (6-column aggregate, Platnick et al. (2017)), where a gamma-size distribution is assumed. Recall that sub-mm has similar penetration depth with passive visible spectrum (Fig. 1b). Such an inconsistency could therefore only be explained by the inconsistent PSD assumptions, or the fact that IceCube and MODIS visible band are observing different parts of the ice particle size spectrum. Another discernible discrepancy is from the simulation using DARDAR microphysics (blue dashed line in Fig. 7). DARDAR is a joint CloudSat radar-CALIPSO lidar IWC retrieval product which is believed to provide the best global knowledge about IWC vertical structures by far. However, with DARDAR spheroid assumption, neither of the PSDs could reproduce IceCube ice observations in the upper tropopshere. This strongly indicates that "one-size-fits-all" does not work for all cloud ice particles. Aggregates and aggregate mix seem to overall produce the best match to the IceCube observations. Spread of the simulated PDF lines occur at warm $TB$ ($\sim 220K$), which indicates that sub-mm channel is more sensitive to particular shape than MW channels (e.g., similar simulations carried for 190 GHz in Ekelund et al. (2020)).

To summarize this section, the comparison between IceCube $TB$ distribution against a variety of ARTS model simulations demonstrate the good quality of IceCube data, especially for the cold ice cloud. The difference on the warm end widths indicates that IceCube data probably contains $\sim 7K$ random noise. On the other hand, spread of the simulations cover the IceCube measurements fairly well, reflecting model's capability at simulating cloud ice scattering at the sub-mm range. This model is serving as the core RTM for the upcoming Ice Cloud Imager (ICI) mission with all channels at sub-mm range. Therefore, IceCube Level-1 data (Gong and Wu (2021)) provides a valuable asset for model validation and testing.

### 3.2 Comparison against other observations

In this section, IceCube $TB$ will be compared against two other independent spaceborne observations at collocated footprints to validate the quality of IceCube cloud radiance measurements. These comparisons will be further compared with previous airborne campaign observations to evaluate the consistency and accuracy of IceCube Level 1 data.

The two independent spaceborne observations are CloudSat 2B-CWC-RO IWC retrieval product (Version 05) and CALIPSO-Imaging Infrared Radiometer (IIR) cloud product (Version 4.20). We chose CloudSat 2B-CWC-RO product instead of DARDAR product that was used in the previous section simply because DARDAR data was not available for the IceCube flight period at the time when this part of research was conducted. IIR was chosen because of two considerations. First, it is a passive IR radiometer, so we can scrutinize the sensitivity window overlaps between sub-mm and IR techniques on the IWP spectrum. Secondly, IIR has $1km^2$ footprint size and a $64km \times 64km$ swath and it is geolocated with CALIOP lidar. It has three medium-IR broadband channels centered at $8.65\mu m$, $10.6\mu m$ and $12.05\mu m$. IIR's cloud product was retrieved against



collocated CALIOP lidar measurements and then extended to the entire swath. Therefore, it has the advantage of high spatial
resolution, high accuracy, and wider swath than lidar (hence more collocation samples with IceCube). Details of IIR cloud
product retrieval algorithms and evaluation can be found in Garnier et al. (2013) and Garnier et al. (2020).

Collocated observations between CloudSat-, IIR- and IceCube $TB$ observations within $\pm30°$ view-angle are identified
globally and mapped on Fig. 8c as blue filled circles and red triangles, respectively. The collocation criteria are defined as
time difference within 10 minutes and distance within $30km$. We can see collocation samples are too limited to make a robust
statistics. However, they are enough to put together a steep and near-linear $TB - IWP$ relationship that agrees with ARTS
model simulation on a broad sense (Fig. 8a, model simulation is taken from the one on Fig. 1a), although model simulation
seems to overestimate the $TB$ depression at larger IWP. Note that the partial column integrated IWC profile in the middle
to upper troposphere ($7 - 15km$), or $pIWP$, computed from CloudSat retrievals have been presented and compared here
to account for the maximum penetration depth of IceCube observations in the mid-latitude according to a mid-latitude case
simulation in Gruntzun et al. (2018). We have tried with different bottom heights for the $pIWP$ integration, and found the
slope barely changed. We can also see from the limited collocated samples that IceCube only starts to show sensitivity to
$IWP$ when it exceeds $40g/m^2$, which agrees beautifully with the RTM simulation even though this simulation has quite some
oversimplified settings.

Moreover, IceCube $TB$ is found to show some sensitivity to the equivalent sphere diameter $D_e$ between $50 - 100\mu m$ range
before the slope becomes chaotic (Fig. 8b). We will revisit this point in the comparison with field campaign below.

So far the only other two passive sensors that carry a 874-883 GHz channel are ESA's airborne International SubMil-
limeter Airborne Radiometer (ISMAR) and NASA's airborne Compact Scanning Submillimeter-wave Imaging Radiometer
(CoSSIR). ISMAR only added the 874 GHz channel in the recent flights so data are not publicly available at this moment
(Hammar et al. (2018), Fox (2020)). CoSSIR channel frequencies range from $183 \pm 1, \pm3, \pm7$ GHz (water vapor profiling),
$220, 380 \pm 1, \pm2, \pm3, \pm6$ GHz (temperature profiling), 640 GHz vertically-polarized and horizontally-polarized pairs, to 874
GHz. Designed with both conical and cross-track scan patterns, CoSSIR was onboard the NASA ER-2 aircraft and deployed to
the TC4 (Tropical Composition, Cloud and Climate Coupling) campaign together with a 94 GHz Cloud Radar System (CRS)
(Zhang and Monosmith (2008)). Comparison between CoSSIR 874 GHz and CRS retrievals is equivalent to comparison be-
tween IceCube-CloudSat, except the in-flight comparison should yield a much cleaner and more robust relationship because of
the perfect collocation, lower noise-level and fortune to sample deep convective systems during the TC4 campaign. Evans et
al. (2012) and Davis et al. (2010) described detailed retrieval procedures for $D_{me}$ and $IWC$ from CoSSIR. $D_{me}$ is the mass
weighted equivalent sphere diameter, defined as:

$$D_{me} = \frac{\int N(D_e)D_e^3 D_e dD_e}{\int N(D_e)D_e^3 dD_e} \qquad (2)$$

where $N$ is the number density. Evans et al. (2012) has proven that a dedicated suite of MW and sub-mm cloud radiometer
such as CoSSIR can achieve the vertical profiling of cloud ice. Gong and Wu (2017) demonstrated the usefulness of polarized
640 GHz radiance measurement on inferring ice crystal shape and orientation.

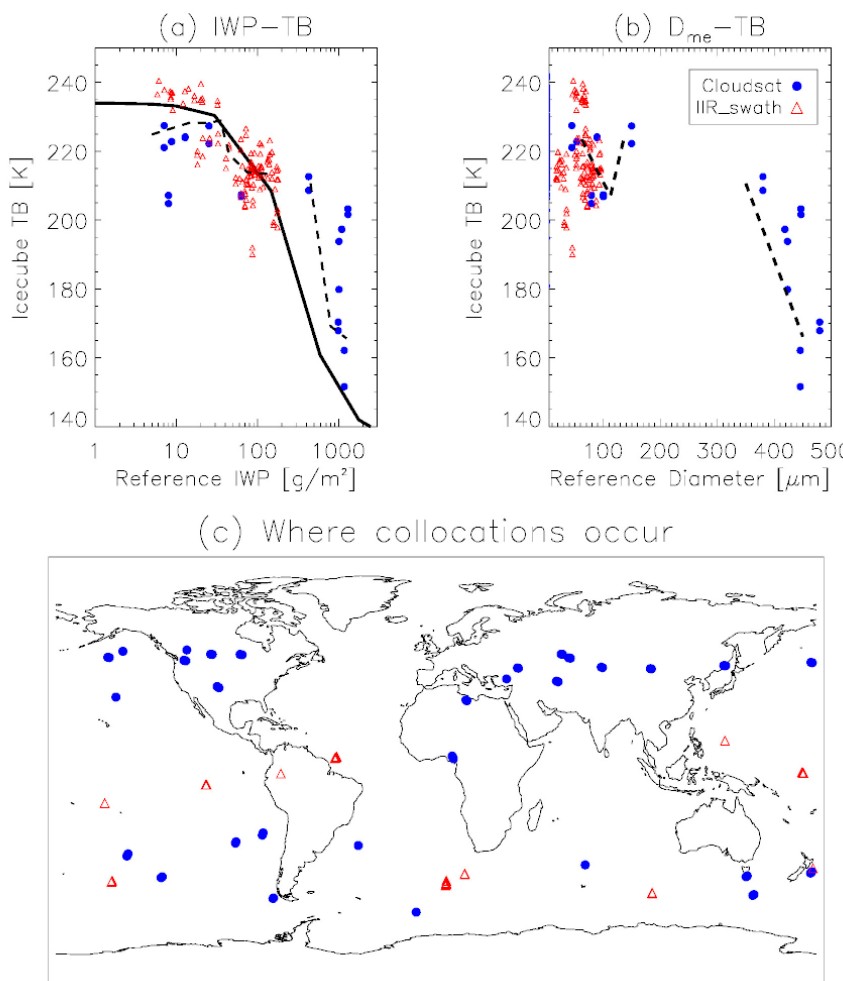

**Figure 8.** IceCube TB show general good correlations with CloudSat (blue filled circle) and CALIPSO-IIR (red open triangle) retrieved IWP (left) and $D_{me}$ (middle). ARTS model simulated $TB - IWP$ relationship (the black solid line from Fig. 1a) is overlaid in the left panel. Dashed lines connect the mean of every $IWP$ or $D_e$ bin, where the bin size is determined on log-scale for $IWP$ and linear scale for $D_e$. Note that CloudSat $IWP$ is calculated from 2B-CWC-RO V05 IWC product and integrated between 7-15 km.

Collocated CoSSIR 874 GHz $TB$ - CRS $IWP$ and $D_{me}$ are scattered in Fig. 9. To make fair comparison, only cross-track scan samples with view-angle $< 30°$ are used. Also, again the partial column integration is used for $IWP$ and $D_{me}$ calculation, but for $9km$ above instead of $7km$ above for CloudSat $pIWP$ integration. This consideration is to account for the tropopause height difference between mid-latitude and deep tropics, because almost all CloudSat-IceCube collocations happen at mid-latitude while TC4 campaign flew in deep tropics. Comparisons from IceCube-CloudSat and IceCube-IIR are overlaid as blue filled circles, and difference between physical meanings of $D_{me}$ and $D_e$ is ignored here. To account for gas absorption between the flight altitude and satellite altitude, a $10K$ warm offset is added to the spaceborne comparison data. From visual inspection

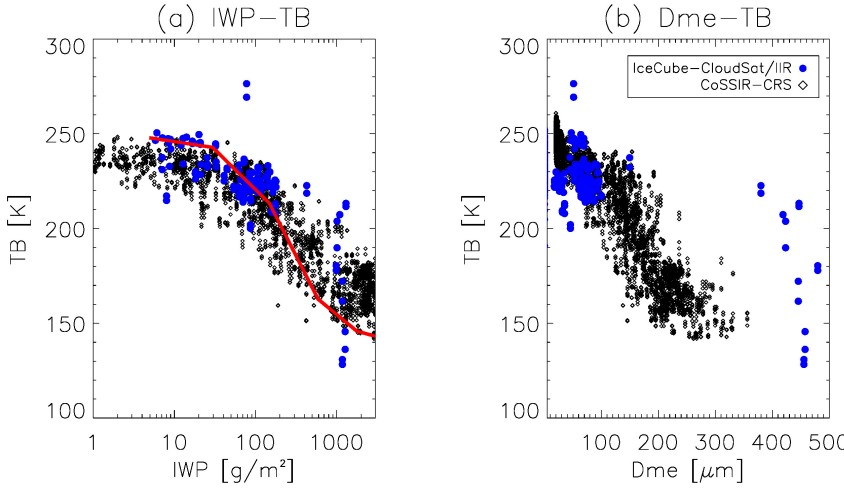

**Figure 9.** (a) $IWP - TB$ and (b) $D_{me} - TB$ relationship derived from collocated CoSSIR-CRS measurements (black dots) during the TC4 campaign. Note that $IWP$ here is the partial column integration from 9km above. IceCube-CloudSat/IIR data from Fig. 8 are overlaid as blue dots with a warm offset of 10K added. ARTS simulation (red curve) is added as the reference.

we can tell the $TB - IWP$ and $TB - D_{me}$ relationships are robust and consistent with those spaceborne relationships. The

scattered spaceborne observations at $\sim 1000 g/m^2$ and $400 - 500 \mu m$ are from one collocation overpass at central U.S. While it is reasonable to argue that these may be outliers or may caused by imperfect collocations, their spread and mean values indicate the upper boundary of 874 GHz sensitivity thresholds, as also suggested by the flattening end to the right of Fig. 9 from the campaign data.

     In summary, $TB - IWP$ and $TB - D_e$ relationships observed by collocated IceCube and other independent spaceborne

observations show impressively good agreement with RTM prediction as well as airborne observations. Based on these two sets of comparison, we can conclude that IceCube observations can indeed fill-in the sensitivity gap between passive IR and MW sensors for $IWP$ and ice particle size. CoSSIR is anticipated to be deployed again in the near future for the IMPACTS (Investigation of Microphysics and Precipitation for Atlantic Coast-Threatening Snowstorms) campaign, where more 874 GHz measurements are going to be collected from mid-latitude winter frontal systems. IceCube data (Gong and Wu (2021)) can

be used as observational references for future RTM simulations or feasibility studies for developing new sub-mm instruments (e.g., CoSSIR, ICI).

# 4   Cloud ice science

IceCube provides the first-ever global ice cloud observation at 874-883 GHz. Although IceCube was not designed as a science mission, it is still worthwhile exploring and discussing the scientific values of this dataset (Gong and Wu (2021)). We don't

intend to provide the Level 2 retrievals at this moment mainly due to two considerations. Firstly, the geolocation registration





has $\sim 14 km$ uncertainty as aforementioned. As cloud can often be very inhomogeneous, pixel-by-pixel retrieval might yield poor quality, especially at large view-angles. Albeit, we will show in the last example that IceCube data can be used for weather-scale study as well. Secondly, with only one single-frequency channel, the retrieval cannot avoid large uncertainties induced by microphysics assumptions. Even it is backed-up by sophisticated RTM simulations, one can clearly see from Fig. 7

how different microphysics assumptions can make huge difference in the simulated $TB$ and therefore impact the retrieved $IWP$. Nevertheless, IceCube data (Gong and Wu (2021)) are suitable for climatology studies. The IceCube observed cloud ice distribution and diurnal variation can help reveal the physical processes that were missing in the IR and MW pictures.

To avoid arbitrarily setting too many assumptions before using a RTM (e.g., ARTS) for retrieval, we applied the empirical relationships of $TB - pIWP$ and $TB - D_{me}$ derived from the airborne campaign CoSSIR-CRS collocation statistics shown

in Fig. 9. A 10K gas absorption offset is added to IceCube $TB$ before the conversion. Apparently a positive value of $IWP$ or $D_{me}$ should not be assigned for every $TB$ observation. Accounting for the sensitivity range and general statistics, a commonly used cloud-screen method, called iterative $3\sigma$ method, is used. This method and the empirical retrieval approach has been used previously in Gong and Wu (2014) and Wu et al. (2014). A 10-loop iteration is carried out to screen out the clear-sky $TB$. In each iteration, the standard deviation $\sigma$ and the peak value corresponded $TB_{peak}$ are calculated, and then any $TB$ values below

the $TB_{peak} - 2\sigma$ threshold are excluded from the next iteration step. After several iterations, the contribution from the long left-tail of the $TB$'s PDF to the skewness can be removed, and the Gaussian spread of the clear-sky variability can be relatively well captured. We then apply $TB_{peak} - 3\sigma$ from the last iteration step as the threshold to separate clear-sky and cloudy-sky scenes. Only pixels with $TB < TB_{peak} - 3\sigma$ are used for the "retrieval". This procedure has been carried out monthly for every $5°$ latitude band, and only for observations at view-angles $< 30°$.

Fig. 10 shows the geographic distribution of mean $IWP$, cloud occurrence frequency (OF), and mean $D_{me}$ for June-July-August, 2017 and 2018 (left), and January-February, 2018 (right). December 2017 was not included because of too few data samples collected during that month. Note that by saying $IWP$, it is actually the partial column integrated $pIWP$ from about $9 km$ above in the tropics and $7 km$ above in the mid-latitude based on the empirical relationships. As expected, large $IWP$, OF and $D_{me}$ are identified in the tropical deep convective regions. The geographic distributions of IceCube $IWP$ agree well with

Aura-Microwave Limb Sounder (MLS) 640 GHz retrieved $pIWP$ above 10 km (Wang et al. (2021)), Odin-SMR (501 and 554 GHz) and SMILES (624-650 GHz) retrieved $pIWP$ above 260 hPa (Eriksson et al. (2014)). Overall the magnitude of IceCube $IWP$ is slightly larger than the above three other passive satellite measurements, but within good agreement with CloudSat $pIWP$ above 260 hPa (Eriksson et al. (2014)). This is more or less expected because we used the empirical relationship, while Odin-SMR and SMILES $pIWP$ retrievals are based on RTM simulations, while Aura-MLS 640 GHz retrieval is based

on the emprical relationship against CALIPSO lidar. OF derived from IceCube data ranges between $15 - 30\%$ in the tropical deep convective regions. Comparing to MODIS IR band retrieved ice cloud coverage at $40 - 80\%$ in the same regions which is more widely spread (Wang et al. (2016)), and to CloudSat observed deep convective clouds occur at a frequency of $5 - 20\%$ in the deep tropics and more narrowly defined (Sassen et al. (2009)), we can thus reconfirm that sub-mm observed cloud is in the middle of the developing processes of deep convective systems between deep convective cloud and IR observed cirrus

and anvils when they spread out, as well as in the middle of the decaying processes when the anvil and cirrus clouds settle



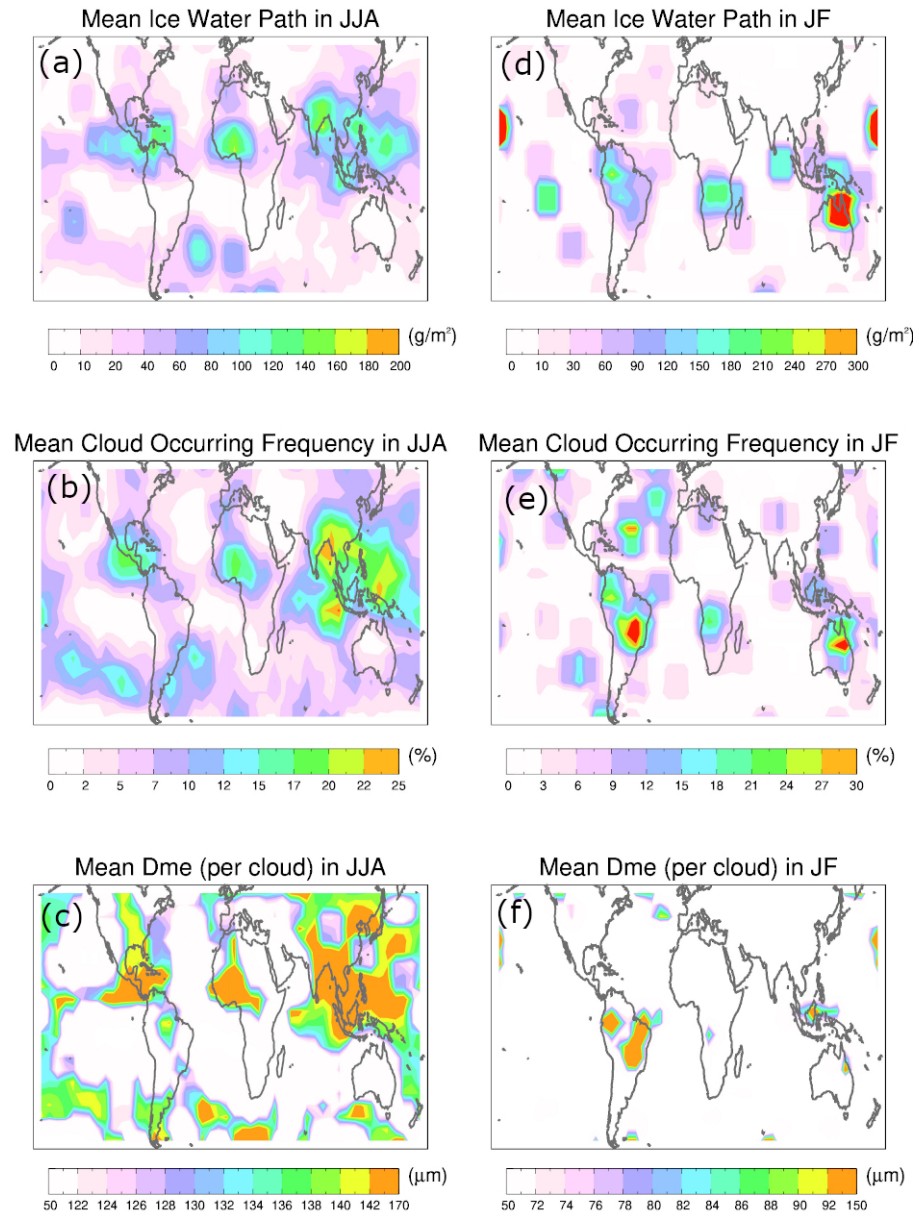

**Figure 10.** Global distribution ($55°S-55°N$) of IceCube $pIWP$ (top), cloud occurrence frequency (middle) and cloud mean $D_{me}$ (bottom) for June-July-August, 2017 and 2018 (left) and January-February, 2018 (right). Grid box size is $5°$ latitude $\times$ $7.5°$ longitude.

down. Mean $D_{me}$ however, starts to loose its geographic differentiation in these regions, indicating that it means the sensitivity threshold in deep convective regions. Other than tropical deep convective regions that migrate with the season shift, coherent



enhancements of the three parameters are also found at the Southern ocean storm track and cold air outbreak regions in austral winter and North Atlantic storm track regions in boreal winter.

As we argued about the importance and unique merit of sub-mm techniques, it is more straightforward to compare the diurnal cycle from sub-mm observations with surface precipitation to identify some plausible coupling processes between the cloud and precipitation. Fig. 11 overlaps the diurnal cycle of IceCube cloud $IWP$ with that of the surface precipitation rate (PRS) retrieved by Global Precipitation Mission - Dual Precipitation Radar (GPM-DPR) in the deep tropics ($20°S - 20°N$) for ocean and land conditions, separately. Over tropical land, we can see IceCube cloud diurnal cycle lags the minima and

maxima of surface precipitation by $\sim 3 - 5$ hours. Early evening heavy precipitation at around 5 PM is well-known as a result of the development of tropical mesoscale convective systems (MCSs) (Nesbitt and Zipser (2003)). While deep convective systems downpour, its upper level cloud continues to develop and spread out. IceCube cloud $IWP$ peak timing (10 PM) strongly suggests they are likely anvils or thick cirrus. As land convection calms down and eventually dissipate and surface precipitation reaches minimum at $\sim 10$ AM, the upper-level cloud does not completely dissipate until noon. So from the diurnal

cycles of precipitation and cloud over land, we may conclude that these IceCube observed clouds are likely slaves of those deep convective clouds and cloud systems.

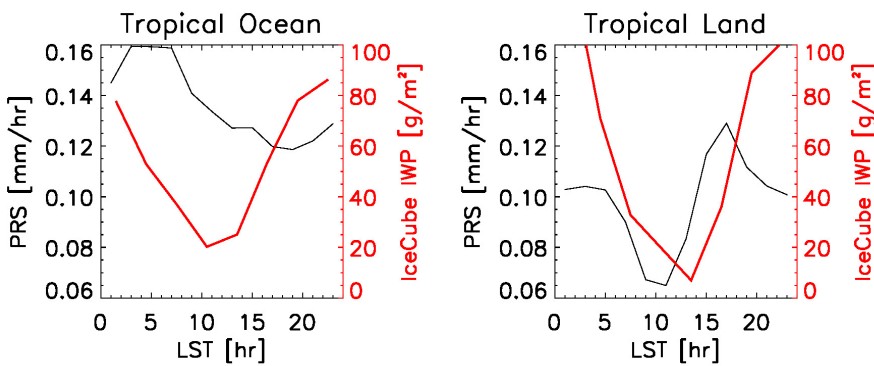

**Figure 11.** Diurnal cycle for IceCube $IWP$ (red) and surface precipitation rate (PRS) derived from GPM-DPR radar (black) over tropical ocean (left) and land (right) with respect to local solar time (LST). Only IceCube data within $\pm30°$ view-angle are used for generating this statistics and tropics is bounded by $20°S$ and $20°N$. Time interval is every 3 hours, and IceCube does not have samples between 10 PM to 1 AM.

    However, the diurnal cycle of precipitation and IceCube cloud over tropical ocean tell a different story. Firstly, the magnitude of diurnal cycle of oceanic precipitation is significantly smaller than that over tropical land although the mean is larger. The overall precipitation peak at 5 AM is believed to be a mixed signature among isolated convection, shallow convection and

MCSs (Nesbitt and Zipser (2003)). However, IceCube observed ice cloud leads the development of surface precipitation by about 2-3 hours, so does the trough (i.e., dissipation phase). This gives us some hint that stratiform precipitation from top down is likely the dominate physical processes rather than bottom-up convective precipitation in determining the surface precipitation diurnal cycle over the tropical ocean.

There are some caveats about this diurnal comparison though. IceCube does not collect enough cloudy-sky samples between
10PM and 1AM in the tropics, so the diurnal cycle is not complete. Due to the same reason, we cannot further scrutinize
different regions (e.g., marinetime continent) so different mechanisms are unavoidably mixed together. Moreover, IceCube
diurnal cycle over tropical land is similar to that derived from SMILES, but its diurnal cycle over the tropical ocean is too
strong compared to that of SMILES (Millan et al. (2013), Eriksson et al. (2014)). The discrepancy between the two passive
sub-mm missions could not be understood without further sub-mm missions that scan the Earth at different local times. Multi-
channels will greatly improve the retrieval quality and increase the retrievable microphysical parameters (Eriksson et al.
(2020)).

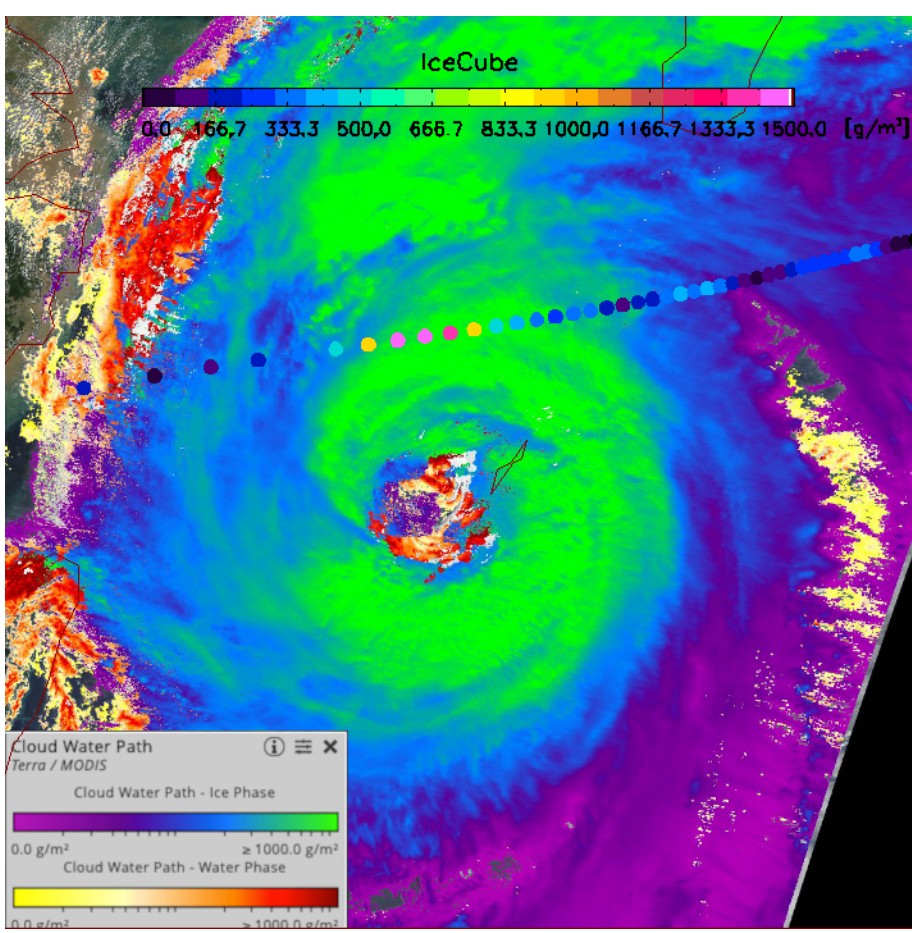

**Figure 12.** $IWP$ retrievals of Typhoon Trami on September 29, 2018 from an overpass of IceCube (colorbar on the top) and Terra-
MODIS (colorbar on the bottom left corner). Their time difference is about 1.5 hours. MODIS retrieval image is grabbed from
https://worldview.earthdata.nasa.gov.

In the last example to demonstrate IceCube data's scientific merit, we show a case study of Typhoon Trami (Fig. 12). IceCube
overpassed this typhoon on September 29, 2018 at an orbital altitude of $\sim 250km$, 4 days before it re-entered the Earth's





atmosphere. However, it was still capable to yield a set of scientifically valuable observations in terms of both the retrieved

IWP value range compared with Terra-MODIS retrievals, as well as the geolocations. Although there is a $\sim 1.5 hrs$ time difference between the Terra-MODIS and IceCube measurements, both observations exhibit good agreements with each other: large IWP values ($> 1000 g/m^2$) for the north arm, and medium IWP values ($200-500 g/m^2$) for the outer bands. IceCube data shows sharp gradient of retrieved IWP from left to right across the north arm while MODIS visible band apparently saturated and cannot tell more detailed structures within the band.

## 5 Conclusions

IceCube carries the first-ever spaceborne 874-883GHz radiometer, which kept on aquiring Earth's ice cloud measurements for more than 15 months. In this paper, we discussed the motivation and algorithms to obtain IceCube Level 1 radiance data (Gong and Wu (2021)). The detailed procedures for data processing have been documented in Section 2. The main steps include space counts prediction and calibration; view-angle determination and geolocation registration; and gain model construction.

The processed IceCube radiance data are then compared with RTM simulated clear-sky radiances, collocated active and passive satellite observations and airborne campaign data to validate its quality. Overall IceCube Level 1 data are found of good quality at near-nadir view-angles. Data quality in general decreases in 2018 compared with 2017 because of instrument degradation. The estimated uncertainty of IceCube radiance data is $\sim 7K$.

Scientific values of IceCube data are discussed with three examples presented. The first is the global 883 GHz cloud ice

map. The agreement and disagreement with what have been found from other passive and active spaceborne measurements demonstrate unique asset of this dataset in filling the missing piece of the entire coupled cloud-precipitation process. Then we show the diurnal cycle to further demonstrate this point. A typhoon case is given at last to showcase that IceCube data are not only unique and important for understanding the climatologies, but also valuable for weather scale studies.

A few sub-mm sensors will be launched to space hopefully in the upcoming years. Together with their airborne and ground

variants as well as their predecessors such like IceCube, we will gain more comprehensive understanding of this band and exploring more capabilities from this band for better monitoring and predicting Earth's weather and climate.

## 6 Code availability

Data processing codes are available upon request.

## 7 Data availability

IceCube Level 1 data is available at NASA open data repository at https://doi.org/10.25966/3d2p-f515. It is also available to the public on IceCube main website at: https://earth.gsfc.nasa.gov/climate/missions/icecube/. Variable name list and meaning can be found in the Appendix E.





## Appendix A:  Details of Fig. 1a

As CloudSat, CALIPSO and AIRS are the spaceborne radar, lidar and passive infrared sensor flying on the A-train constellation,
only one-month of tropical collocation statistics (January 2009, $30°S - 30°N$) is robust enough. The collocation criteria is
defined such that the spatial difference should be less than 5 km, and temporal difference should be less than 1 minute. AIRS
water vapor channel #1247 ($wavenumber = 1128.57cm^{-1}$) is employed here. AIRS $T_{cir}$ is computed by subtracting Level-2
cloud-cleared radiance ($T_{ccr}$, Version 6) from Level-1B brightness temperature $TB$ (Version 5). CloudSat-CALIPSO jointed
IWP retrieval product 2C-ICE (Version 4) is used as the "truth" showing on the horizontal axis. In the case that multiple
CloudSat-CALIPSO footprints collocated with one AIRS footprint, the IWP values retrieved from the former are averaged first
before constructing the 2D-PDF. The blue solid line connects the peaks of AIRS-CloudSat/CALIPSO 2D PDF.

To construct MHS's $T_{cir} - IWP$ relationship, collocated tropical near-nadir samples from multiple months are used to
compile the statistics. Details about collocation criteria, near-nadir definition, etc., can be found in Gong and Wu  (2014).
Different from above blue line, the black dashed line is from an ARTS model simulation at 190 GHz by inserting a Gaussian
shape of ice cloud layer between 150 hPa and 600 hPa with the peak at 400 hPa. One can see the RTM simulation, even though
using an idealized cloud layer, can accurately mimic the observed $T_{cir} - IWP$ relationship, indicating that MW channels are
more sensitive to the total mass than to the vertical structure.

The black solid line is based on another ARTS model simulation at 874 GHz with the same ice cloud configuration. For both
the simulations at 190 and 874 GHz, $T_{cir}$ is derived by subtracting the simulated clear-sky radiance (i.e., setting $IWC = 0$ in
the entire column), so any surface contribution is excluded using this strategy.

## Appendix B:  Poor-quality orbit examples

Fig. 2 and Fig. 3 only give an example of good measurements. In the later time of the IceCube mission, various experiments
were tested to gain knowledge of instrument behavior under different scenarios. Fig. B1 and Fig. B2 here gave an example of a
"poor quality" orbit. IceCube was kept on for several orbits actually, but the instrument temperature was kept under $35°C$. As
a result, one can clearly see the oscillation in the later part of this orbit, and the $C_e$-$C_{sp}$ contrast declines over time due to the
relatively high temperature (Fig. B1a). In Fig. B1d, it is also clear that the periodic low-frequency oscillation of $C_{sp}$ against
$Tp_3$ and $Tp_4$ also changed after $Tp_4$ hit the temperature cap and IceCube was cooled down and then heated up again. As a
result, Fig. B1c and Fig. B1d failed to capture the slowly-varying oscillations of $C_{sp}$. Therefore, we need to further apply a
fitting to $C_{sp}$ (green line in Fig. B2) and the final $C_e$ now remains largely stable among different spins (red dots of Fig. B2,
shifted upward by 200 counts). Some spins that have too low $C_e$-$C_{sp}$ contrasts (e.g., the 5 spins between $dt = 1.0E + 4s$ and
$dt = 1.1E + 4s$) or too slow spin velocity (e.g., the last 4 spins) are excluded.

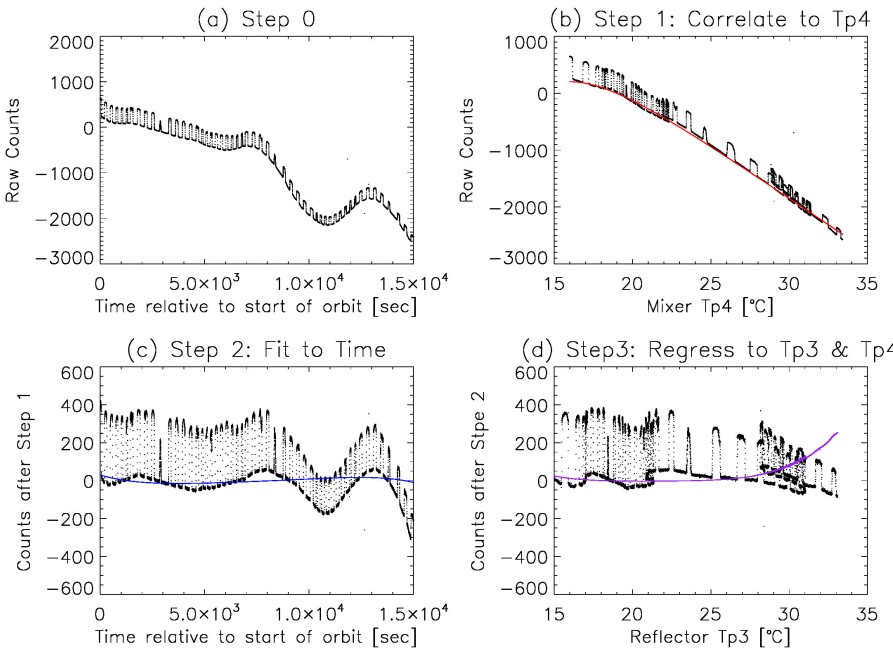

**Figure B1.** Same with Fig. 2, except this is an example showing a bad-quality orbit. This orbit is the second orbit of IceCube on April 2, 2018.

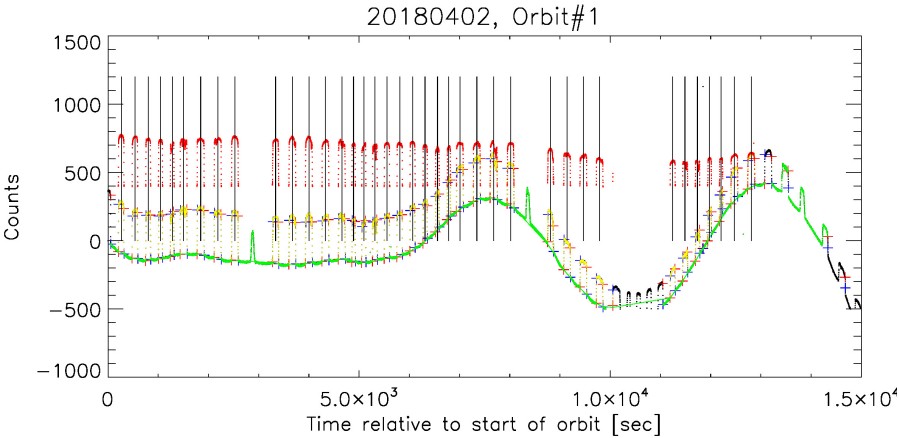

**Figure B2.** Same with Fig. 3, except this is the final step for the case in Fig. B1. The final polynomial fitting to the space-view counts only is a very necessary step to capture the slowly varying component of the timeseries, which was failed to be captured in the Step 3 based on Fig. B1c using all-sky counts.



**Appendix C:  Machine Learning/Artificial Intelligence Model**

Empirical relationships have been used in Steps 1-4 to account for the majority of $C_{sp}$ variations even for irregular orbit like the one shown in Fig. B2. Therefore, the residual between observed $C_{sp}$ and predicted ones remain stable with a standard
deviation of $\sim 4K$ (black crosses in Fig. 4). During these procedures, instrument parameters such like spin rate ($spin_x$, $spin_y$ and $spin_z$) , measured magnet field ($Mag_x$,$Mag_y$,$Mag_z$) were not used. They were believed to not affect $C_{sp}$ in the pre-launch tests. As such, ML/AI model is only included in this appendix section for the purpose of testing the robustness of this approach in reducing any generic instrument noise for IceCube and future cubesat type of missions or constellations that are less well calibrated due to the low cost-cap. In the final product, we provide both the TB before and after this ML/AI step for
user to select.

Only the random forest model was tested for this work. As we can see from the "poor orbit" case in Fig. B1b and B1d, $C_{sp}$ shows a hint of slight elevation after IceCube has been switched on for a long time given the same $Tp_3$ and $Tp_4$. Therefore, relative change of time and temperature with respect to the switching-on parameters for each orbit are also factored in as $DT_1$, $DT_2$, $DT_3$, $DT_4$ and $DTime$. The Julian day counted from January 01, 2017 is marked as "$Time$". As there's no need or
intention to "predict for the future" but rather to capture the fast-varying and slow-varying components of the $C_{sp}$ residual, we split the total samples randomly so to assign 70% for training and the rest 30% for testing/validation. This is different from traditional ML/AI model performance check, and leave some caveat for argument admittedly. Only the 30% testing sample statistics are shown in Fig. C1.

As one can see from the heatmap in Fig. C1a, the majority of the predicted residual centered around $0K$, meaning that
the trained model can largely capture the residual variations most of the time (recall "residual" is defined as the discrepancies between the predicted and the observed $C_{sp}$ values). It also indicates that there's no associated bias to skew the statistics. Standard deviation of the predicted residual $\sigma_{sp}$ is $\sim 2K$ as opposed to $\sim 4K$ before the ML/AI treatment, which is a direct reflection of the strength of this ML/AI approach in capturing the sophisticated, "white-noise-alike" features of the residuals. In the meantime, large residual values up to $\pm 10K$ can be produced as well, showcasing along the 1:1 line in the heatmap.
These values mainly occur at the start and end of the Earth-view leg with big slant-view angles.

Rank of importance among all variables can help further diagnose the sources of the residual time series. As expected, instrument degradation (i.e., $Time$) and switch-on time duration affect the residual noise of $C_{sp}$ the most. In addition, the relative change of $Tp_4$ and $Tp_3$, indicated by $DT4$ and $DT3$, as well as the spin rate along three axes contribute about the same amount of importance to the final prediction.

**Appendix D:  $TB$ uncertainty**

The uncertainty (i.e., error bar) of derived $TB$ can be calculated as follows. According to the definition of $TB$, $TB = C_e/G$, therefore,

$$dTB = \frac{\partial G}{G} \cdot \frac{C_e}{G} + \frac{\partial C_e}{G} = \frac{\Delta G}{G} \cdot TB + \frac{\Delta C}{G} \tag{D1}$$

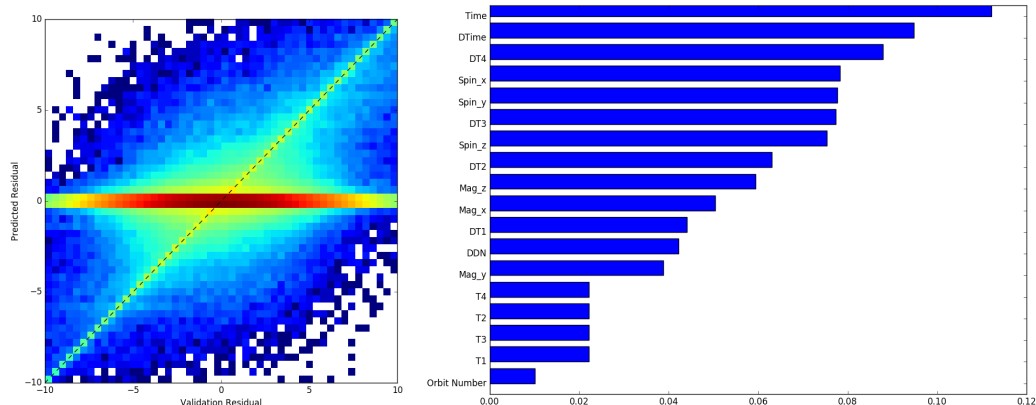

**Figure C1.** (left) Heatmap of the predicted $C_{sp}$ residuals (vertical axis) against the observed ones (horizontal axis), colored in log-scale. (right) Rank of importance of different parameters.

The first term is uncertainty induced by Gain estimation, and the second term is induced by the space count prediction. $\Delta C$ is

calculated for each orbit as shown in Fig. 4, which should be $\sim 4K$ ($\sim 2K$) before (after) the ML/AI process. $\Delta G$ is computed from the standard deviation of PDFs of daily G, the value of which varies around 0.002 - 0.005 [counts/K] for most days, and occasionally reach up to 0.02 [counts/K]. $dTB$ is thus calculated for each given $C_e$ once $TB$ is converted. This value is reported in the L1 data as well.

**Appendix E:  Variable name list**

IceCube Level 1 data (Gong and Wu (2021)) are stored in HDF5 format. The data file name is IceCube.L1.YYYYMMDD.V01.h5, where YYYYMMDD indicates the year (4-digits), month (2-digits) and day (2-digits).

The variable name list can be found below:

LAT - Latitude; Unit: [degree]

LNG - Longitude; Unit: [degree]

TB_MODEL - RTM simulated clear-sky $TB$; Unit: [K]

TB_OBS1 - Observed $TB$; Unit [K]

TB_OBS2 - Observed $TB$ with ML/AI predicted residual subtracted. See the Appendix C for detail of this procedure. Unit: [K]

TB_UNC1 - $TB$ uncertainty; Unit [K]

TB_UNC2 - $TB$ uncertainty after ML/AI treatment; Unit [K]

UTC - Universal time; Unit: [second]

VIEW_ANG - View angle from the nadir; Unit: [degree]

DN_FLAG - Day/Night flag (0: Day; 1: Night); Unitless.



QC - Quality control flag; Unitless.

0: Good quality

1: Geolocation quality is poor due to large view-angle

2: Quality is doubtable due to abnormal gain

3: Quality is doubtable due to abnormal spin rate or instrument temperature

ORBIT_NUMBER - Each orbit number is counted from the switch-on time to the switch-off time. Everyday this variable

resets to 0. Unitless.

*Author contributions.* DW provided the original thoughts and procedures for data calibration and processing. JG refined and improved the algorithm, conducted the data analysis and wrote the manuscript. PE provided the ARTS model simulation. All authors were heavily involved in interpreting the results.

*Competing interests.* The authors declare no conflict of interest.

*Acknowledgements.* This work is supported by NASA's Grant NNH19ZDA001N-RRNES. The authors are grateful to the entire IceCube team for the successful fabrication and launching of the IceCube instrument as well as industrious effort on the data collection. NASA Earth Science and Technology Office (ESTO) is acknowledged for the selection and support of IceCube mission. Dr. Yuping Liu contributed to the early data processing and $T_{ccr}$ computation, whose work is indispensable to the completion of the current paper. Helpful discussions with Drs. Stuart Fox, Anne Garnier, Inderpreet Kaur and Chenxi Wang are greatly appreciated.



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
