# Peer review of "The first global 883 GHz cloud ice survey: IceCube Level 1 data calibration, processing and analysis"

_Earth System Science Data, 2021_

## Author Comment (AC2)

ESSD-2021-101 final response:

RC1: This study presents the first global 883 GHz cloud ice survey, which is important for Earth's climate and weather (e.g., solid precipitation). The IceCube Level 1 data calibration, processing and analysis have been described in detail. Although I am not an expert in this speical research field and could not provide careful comments on details, I believe that this study is a good step in observing, quantifying and understanding the cloud ice globally.

Thanks for your kind words and acknowledgement of our work and its importance. Based on your and the other reviewers' suggestions, we've included one more Appendix section (now Appendix E) to thoroughly discuss IceCube instrument noise estimation. We also rewrite the paragraph around Line 365 to clarify the different cloud-precipitation processes over ocean and land that we can derive from the diurnal cycles of IWP and surface precipitation. Hope the revision can enhance the quality of the presentation.

RC2: This paper describes the detailed procedures for IceCube Level 1 data calibration, processing and validation, and the scientific values of the data. IceCube provides the first global ice cloud observation at 874-883 GHz, which is a critical dataset to the satellite and science communities. The manuscript is relatively well structured, even though the presentation still has room for improvement. I only have some minor questions and comments.

Thanks for your kind words and acknowledgement of our work. Per your suggestions, we've included one more appendix section (now Appendix E) to thoroughly discuss IceCube instrument noise estimation. We also rewrite the paragraph around Line 365 to clarify the different cloud-precipitation processes over ocean and land that we can derive from the diurnal cycles of IWP and surface precipitation. Hope the revision can enhance the quality of the presentation.

1. Lines 214-216: Large discrepancies are shown in Figure 7 for TB > 200 K. The authors claim that this is due to the instrument noises. It would be more convincing to readers/me if the comparison was shown between the noises-added simulation and no-noise simulation.

This is an excellent point. We apologize for not showing the supporting figure, which is included below and also in the newly added Appendix E (the original Appendix E is now moved as Appendix F). Line 240 is modified together to point interested readers to read Appendix E closely. In this figure, PDFs from two simulations with no-noise (blue) and a randomly added 7K Gaussian noise (orange) are compared against that from IceCube TB, and we can clearly see the latter (orange) agrees well with the observation on the warm end.

[Figure]

Figure R1: PDF comparison of IceCube

2. Lines 221-224: What is the definition of the spheroid particles in the simulations. I am pretty surprised that this assumption is particularly bad compared to the obs. Smaller ice particles tend to be more sphere or somewhat spheroid, but it's odd to me that there are such differences between the two simulations. What could be the reason for that?

The entire ARTS simulation setting in this paper adopts exactly the same architecture and model of Ekelund et al. (2020), which is cited in the reference list. For example, "*soft-spheroid is defined as a spheroid composed of an air-ice mixture*", and the DARDAR spheroids "*are oblate with an aspect ratio (ratio of the minor to the major axis) of 0.6 and follow the mass–size relationship given in Delanoë et al. (2014, Eqs. 13–15).*"

In that paper, the authors found that spheroids were often used for active and passive remote sensing due to its computational efficiency. "*Such configurations can yield good results at single frequencies, for instance, through fine-tuning of the particle effective density (Galligani et al., 2015). However, they fail to provide consistent results at multiple frequencies and are not appropriate for multi-frequency measurements and combined passive–active applications (Geer and Baordo, 2014).*" One of the final highlights of Ekelund et al. (2020) is that they found MW frequencies (e.g., GPM-GMI) were relatively insensitive to particle shape, while sub-mm frequencies were very sensitive to.

Note that, using completely independent RTMs with completely independently calculated scattering database (Yang et al., 2013), MODIS team identified a hexagon-aggregate habit best fits their visible and infrared observations globally, and hence adopted the hexagon-aggregate habit as their retrieval assumption for Collection 6 (Platnick et al., 2017). CERES and POLDER

also assumed non-spheroid shapes for their retrieval algorithms (Yang et al., 2018, https://doi.org/10.3390/atmos9120499).

In summary, numerous observations (ground, airborne and spaceborne) have identified that cloud ice habits in nature are dominated by non-spherical and non-spheroidal shapes, and satellite retrieval community is on the pathway to abandon such simplified assumptions. Cloud microphysics modeling community is a little behind, I'd say, but efforts are on the way to simulate the nature (e.g., Geer et al., 2021, https://doi.org/10.5194/gmd-2021-73).

3. Lines 364-366: The ocean vs. land comparison is very interesting, but the explanation for the noon IWP minimum over the ocean is not that clear in the text. It states that "stratiform precipitation from top down is likely the dominate physical processes rather than bottom-up convective precipitation in determining the surface precipitation diurnal cycle over the tropical ocean" The stratiform and anvil clouds associated with convection tend to last a bit after the maximum precipitation happens, but we see a decrease in IWP from 5 to 10 LST. Could you comment on that?

Apologize for not writing clearly in this part. What we (mainly the first author) try to speculate here, is that IWP diurnal cycle observed by IceCube is likely driven by the development over anvils from convections over tropical land (hence, a time-lag of ~ 3 hours); but the diurnal cycle of oceanic precipitation is associated with the dissipation of anvils (hence, a lead of time of ~ 6 hours). It takes much less time for the "bottom-up" convective downpours than that of "top-down" stratiform precipitation processes as illustrated below in the conceptual model (Fig. R2). TRMM satellite observations suggest that stratiform precipitation processes dominant the tropical oceanic precipitation, as shown in Fig. R3.

Now this paragraph has been re-written as follows:
*"However, the diurnal cycle of precipitation and IceCube cloud over tropical ocean tell a different story. Firstly, the magnitude of diurnal cycle of oceanic precipitation is significantly smaller than that over tropical land although the mean is larger, which has been reported previously in literature. The overall precipitation peak at 5 AM is believed to be a mixed signature among isolated convection, shallow convection and MCSs (Nesbitt and Zipser (2003)). However, IceCube observed ice cloud leads the development of surface precipitation by about 5 hours, so does the trough (i.e., dissipation phase). This hints that stratiform precipitation forming from anvils is likely the dominate physical process rather than bottom-up convective downpours in determining the diurnal cycle of the tropical oceanic precipitation. This speculation explains the opposite phase-lag between the diurnal cycle of IWP and surface precipitation over tropical ocean versus land. It is also supported by the longer time delay 370 over ocean (~ 5 hours) than that over land (~ 3 hours) as it takes a longer time scale for the stratiform precipitation particle to form from the anvils and to fall down to the ground. Using Tropical Rainfall Measurement Mission (TRMM) products, Yang and Smith (2008) found that stratiform precipitation dominated the tropical oceanic precipitation throughout the day, with more contributions from local afternoon to mid-night. This partially supports our hypothesis. Nevertheless, we could only complete the*

*picture of convection-cloud-precipitation process by wisely using a combination of satellite observations that detect different components of this entire process."*

[Figure]

Figure R2: Conceptual model of particle fountain in a multicellular mesoscale convective system (MCS), adapted from Yuter and Houze (1995, https://doi.org/10.1175/1520-0493(1995)123<1964:TDKAME>2.0.CO;2). One can see the stratiform precipitation (top branch of the fountain) takes longer time in general to fall, while convective downpours happen in a much faster time scale. In our paper here, we call the stratiform precipitation as "top down", while convective core precipitation as the "bottom up" process as it is usually described as the "hot air bubble" model.

[Figure]

Figure R3: The diurnal cycle of Convective (C; thick lines) and Stratiform (S; thin lines) precipitation percentage contribution (%) to the total rainfall rate over tropical ocean (left) and land (right) derived from two TRMM Level 2 datasets. Adapted from Yang and Smith (2008, 10.1175/2008JCLI2096.1). One can see clearly that stratiform precipitation dominates the oceanic precipitation total intensity throughout the day, and its dominancy peaks at local afternoon-early evening.